# Clinical diversity and treatment results in Tegumentary Leishmaniasis: A European clinical report in 459 patients

Romain Guery[1,2]*, Stephen L. Walker[3,4], Gundel Harms[5], Andreas Neumayr[6,7,8], Pieter Van Thiel[9], Jean-Pierre Gangneux[10], Jan Clerinx[11], Sara Karlsson Söbirk[12], Leo Visser[13], Laurence Lachaud[14], Mark Bailey[15], Aldert Bart[16], Christophe Ravel[14], Gert Van der Auwera[11], Johannes Blum[7,8]*, Diana N. Lockwood[17]*, Pierre Buffet[18]*, on behalf of the LeishMan Network and the French Cutaneous Leishmaniasis Study group[¶]

1 Necker-Pasteur Infectiology Centre, Necker-Enfants Malades University Hospital, Institut Imagine, Assistance Publique-Hôpitaux de Paris, Paris, France, 2 Confluent Private Hospital, Nantes, France, 3 Hospital for Tropical Diseases and Department of Dermatology, University College London Hospitals NHS Foundation Trust, London, United Kingdom, 4 Faculty of Infectious and Tropical Diseases, London School of Hygiene and Tropical Medicine, London, United Kingdom, 5 Institute of Tropical Medicine and International Health,Charité–Universitätsmedizin Berlin, corporate member of Freie Universität, Humboldt-Universität zu Berlin, and Berlin Institute of Health, Berlin, Germany, 6 Swiss Tropical and Public Health Institute, Basel, Switzerland, 7 University of Basel, Basel, Switzerland, 8 Department of Public Health and Tropical Medicine, College of Public Health, Medical and Veterinary Sciences, James Cook University, Queensland, Australia, 9 Amsterdam UMC, University of Amsterdam, Department of Infectious Diseases, Amsterdam, The Netherlands, 10 Univ Rennes, CHU Rennes, Inserm, EHESP, Irset (Institut de recherche en santé, environnement et travail)—UMR_S 1085, Rennes, France, 11 Institute of Tropical Medicine, Antwerp, Belgium, 12 Division of Infection Medicine, Department of Clinical Sciences, Lund University, Lund, Sweden, 13 Department of Infectious Diseases, LU-CID, Leiden University Medical Center, Leiden, The Netherlands, 14 Department of Parasitology-Mycology, University of Montpellier, Montpellier University Hospital, French National Reference Center for *Leishmania*, MIVEGEC, Montpellier, France, 15 Birmingham Heartlands Hospital, Birmingham, United Kingdom, 16 Amsterdam UMC, University of Amsterdam, Department of Medical Microbiology, Amsterdam, The Netherlands, 17 Hospital for Tropical Diseases, London School of Hygiene and Tropical Medicine, London, United Kingdom, 18 Université de Paris, UMRs-1134, Inserm, Centre Médical de l'Institut Pasteur, Institut Pasteur, Paris, France

☯ These authors contributed equally to this work.
¶ Membership of the LeishMan Network and the French Cutaneous Leishmaniasis Study group is provided in the Acknowledgements section
* romain.guery@groupeconfluent.fr (RG); johannes.blum@swisstph.ch (JB); pierre.buffet@inserm.fr (PB)

**Data Availability Statement:** All relevant data are within the manuscript and its Supporting Information files.

## Abstract

### Background

Cutaneous leishmaniasis (CL) is frequent in travellers and can involve oro-nasal mucosae. Clinical presentation impacts therapeutic management.

### Methodology

Demographic and clinical data from 459 travellers infected in 47 different countries were collected by members of the European LeishMan consortium. The infecting *Leishmania* species was identified in 198 patients.

**Funding:** The authors received no specific funding for this work.

**Competing interests:** The authors have declared that no competing interests exist.

## Principal findings

Compared to Old World CL, New World CL was more frequently ulcerative (75% vs 47%), larger (3 vs 2cm), less frequently facial (17% vs 38%) and less frequently associated with mucosal involvement (2.7% vs 5.3%). Patients with mucosal lesions were older (58 vs 30 years) and more frequently immunocompromised (37% vs 3.5%) compared to patients with only skin lesions. Young adults infected in Latin America with *L. braziliensis* or *L. guyanensis* complex typically had an ulcer of the lower limbs with mucosal involvement in 5.8% of cases. Typically, infections with *L. major* and *L. tropica* acquired in Africa or the Middle East were not associated with mucosal lesions, while infections with *L. infantum*, acquired in Southern Europe resulted in slowly evolving facial lesions with mucosal involvement in 22% of cases. Local or systemic treatments were used in patients with different clinical presentations but resulted in similarly high cure rates (89% vs 86%).

## Conclusion/Significance

CL acquired in *L. infantum*-endemic European and Mediterranean areas displays unexpected high rates of mucosal involvement comparable to those of CL acquired in Latin America, especially in immunocompromised patients. When used as per recommendations, local therapy is associated with high cure rates.

### Author summary

Cutaneous and muco-cutaneous leishmaniasis (CL and MCL) are disfiguring diseases caused by a worldwide distributed parasite called *Leishmania* and its 20 species. Clinical manifestations span a wide continuum from single nodular lesion to disseminated form with mucosal involvement.

No randomized clinical trial has ever been done exclusively in travellers and medical management is poorly evidence-based or based very predominantly on data obtained in endemic countries. Articles and reviews almost invariably propose a dichotomic view, with Old World CL described as a benign disease in contrast to New World CL strongly associated with destructive mucosal lesions.

Our study is the first prospective clinical study providing a detailed description of the clinical presentation and risk of mucosal involvement in CL in several hundreds of patients, with frequent formal identification of the infecting *Leishmania* species. The harmonized data collection in patients infected in many transmission foci worldwide enabled direct comparisons of clinical patterns induced by different *Leishmania* species, and on the outcome following treatment with either local or systemic regimens. The study is based on an international harmonized data collection that allowed a wide capture of parasitologically confirmed cases. In striking contrast with previous assumptions, the study shows that CL acquired in Europe displays unexpected high rates of mucosal involvement comparable to those of CL acquired in Latin America, especially in immunocompromised travellers. It also shows that when used as per recommendations, local therapy is associated with high cure rates.

## Introduction

Ninety-eight countries and 3 territories are considered endemic for leishmaniasis, a vector-born parasitic disease caused by a protozoan from the *Leishmania* genus [1]. While visceral leishmaniasis is a severe disease, cutaneous leishmaniasis (CL), that affects the skin and can involve the mucosa of nose and mouth, is not life-threatening. It causes however disfiguring lesions, scarring and stigma [2]. CL affects 0.7 to 1 million individuals each year and its global burden is increasing [3,4].

According to existing surveillance networks, CL is not infrequent in travellers visiting endemic areas [5]. The absence of mandatory notification in most countries and the self-curing course of a proportion of lesions hamper robust estimation of the burden of CL in travellers. Recent conflicts in the Middle East have been linked to a rise in the incidence of CL in migrants and refugees [6].

Textbook and reference reviews have long proposed a dichotomic view, with Old World CL often described as a benign cutaneous disease in contrast to New World CL strongly associated with destructive mucosal lesions. Thus, assessment of the mucosal involvement in travellers is discussed only for New World CL and almost never for Old World CL in clinical practice. To provide a detailed description of the clinical presentation and risk of mucosal involvement of CL, both directly impacting treatment decisions, we analysed its presentation in travellers from data deposited in a large international database. The harmonized data collection by clinicians attending patients infected in many transmission foci worldwide enables direct comparisons of clinical patterns induced by different *Leishmania* species, and the outcome following treatment with either local or systemic regimens.

## Methods

### Ethics statement

This observational study (DR-2013-041; N˚912650) was approved by the French National Agency regulating data protection (Commission Nationale de l'Informatique et des Libertés). Patients (or their legal representative) provided verbal or written consent according to national regulations for use of anonymized data on clinical findings, treatment received, clinical outcome and laboratory results. No genetic analyses of human DNA were performed.

### Data collection

From 2006 through 2012 data collection was initiated by a French referral network as reported elsewhere [7,8]. A standardized case report form was used for baseline demographical, clinical and biological data and a second case report form was used to collect outcomes, identification of *Leishmania* species and adverse events, at least 42 days after the expert had provided treatment advice to the attending physician. From 2012 through 2019, the analysis was extended to several European countries by teams belonging to the LeishMan network.[9] LeishMan is a multicentre international medical project aiming to improve the management of leishmaniasis through harmonization of medical practices and collection of data in a common system. The variables of the LeishMan database were translated from those of the initial French database. Currently, the consortium gathers 50 experts affiliated to 30 institutions in 11 European countries (Belgium, France, Germany, Italy, Norway, Portugal, Spain, Sweden, Switzerland, the United Kingdom, The Netherlands). Data were pseudonymously and prospectively collected by experts from each institution. The database is hosted by Epiconcept since 2012 and has been certified as "Health Data Host" (ISO standards ISO 27001 and elements of ISO 2000–1

and ISO 27018). Demographic, clinical and biological data were filled in an electronic case report form.

### Definitions

A patient was considered to have cutaneous/mucocutaneous leishmaniasis if she/he had: (1) cutaneous and/or mucosal lesions; (2) laboratory confirmation of the presence of *Leishmania* as follows: presence of amastigotes in smears or tissue sections and/or promastigotes in culture and/or positive molecular testing by Polymerase Chain Reaction (PCR) on a skin sample.

Mucosal leishmaniasis (ML) refers to the presence of mucosal lesion(s) without skin involvement. Muco-cutaneous leishmaniasis (MCL) refers to the simultaneous presence of both mucosal and skin lesions. Mucosal involvement refers to both ML and MCL. Post-Kala Azar Disease (PKDL) was defined as new skin lesions in a patient who had recovered from visceral leishmaniasis. Disseminated cutaneous leishmaniasis was defined as CL with more than 10 lesions in 2 non-contiguous anatomical sites. Immunocompromised patients include patients with at least one of the following treatments or conditions: immunosuppressive therapy (e.g >5mg/day equivalent prednisone during more than 3 months, chemotherapy, methotrexate, monoclonal antibodies or small molecules targeting immune cells or their products [e.g. anti-TNF agents], HIV infection, primary immunodeficiencies). HIV testing was not systematically done and/or collected as variable.

Healing in CL was defined as complete re-epithelialization for an ulcer or disappearance of induration for a papular lesion at least 42 days after treatment start [10]. Treatment regimen and dosage mostly followed national or international guidelines [11–13].

Information regarding the infecting *Leishmania* species was captured in the case report form by the expert based on the molecular identification performed by local laboratories in each centre or by national reference laboratories. Clustering of (sub)species in complexes followed a recent classification of *Leishmania* species [14]. Briefly, *L. infantum* was included in the *L. donovani* Complex, *L. major* was included in the *L. major* Complex. *L. braziliensis* and *L. peruviana* were included in the *L. braziliensis* Complex, and *L. panamensis* and *L. guyanensis* were merged into *L. guyanensis* Complex.

### Patients eligible for local therapy based on clinical parameters

We determined the proportion of patients with CL potentially eligible for a local therapy (eg, paromomycin cream or intralesional antimony + cryotherapy). Criteria for eligibility were as per the LeishMan consensus for Treatment of Cutaneous and Mucosal Leishmaniasis in Travellers and WHO recommendations [12,15]. Criteria were: localized CL, ≤ 4 lesions, ≤ 4cm in largest induration diameter, no immunosuppression, lesion type (papulo-nodular or dry crust or wet crust), lesion site compatible with treatment method (eyelid and peribuccal lesions excluded).

### Statistics

Continuous variables are presented as median [interquartile range] and categorical variables as numbers (frequencies). Categorical variables were compared using chi-square test or Fisher's exact test as appropriate. Continuous variables were compared using Student's t-test or Wilcoxon-Mann-Whitney test as appropriate. A two-tailed p-value $< .05$ was considered statistically significant. Statistical analyses were performed using R software 3.1 version (R Development Core Team, 2008) using GMRC Shiny Stat application developed by CHU de Strasbourg (2017).

## Results

### Geography

A total of 459 patients with CL were recruited into the study cohort between 2006 and 2019, corresponding to 464 infections, i.e., 5 patients had a second episode (**Fig 1**). Patients were included by 10 centres from 7 countries in Europe (**Tables A and B in S1 Table**). There were 279 cases (60%) and 185 cases (40%) acquired in Old World and New World, respectively. The infection had been acquired in 47 different countries (**Fig 2**). Top 3 countries of acquisition were French Guiana (n = 41), Peru (n = 37) and Costa Rica (n = 28) for the New World and Spain (n = 48), Syria (n = 36) and Morocco (n = 34) for the Old World (**Tables C and D in S1 Table**). Countries of acquisition could not be determined in 10 cases (2%) because patients had travelled to multiple endemic countries in a short period of time.

### Demography

The median age of patients in the study cohort was 30 years; 99 infections (21%) occurred in children <16 years. There were marked differences in age distributions between travellers Visiting Friends or Relatives and tourists (**S1 Fig**). The main reasons for travel in children were Visiting Friends or Relatives (57%) or migration (25%) while travelling reasons in adults were more heterogeneous and influenced by travel-destination. For example, tourists returning from the New World (often referred to as "backpackers") were younger than tourists from the Old World (median age 30 years vs 48 years; p = 0.03). Migration-related infections were only observed in the Old World while military personal or expatriates acquired the infection almost exclusively in the New World. Patients were immunocompromised in 5% of cases, and a previous history of leishmaniasis was reported by 8% of patients in the cohort. Compared to those infected in the Old World, patients infected in the New World were more frequently males (75% versus 54.5%), and more rarely immunocompromised (0.6% versus 8%) (**Table 1**).

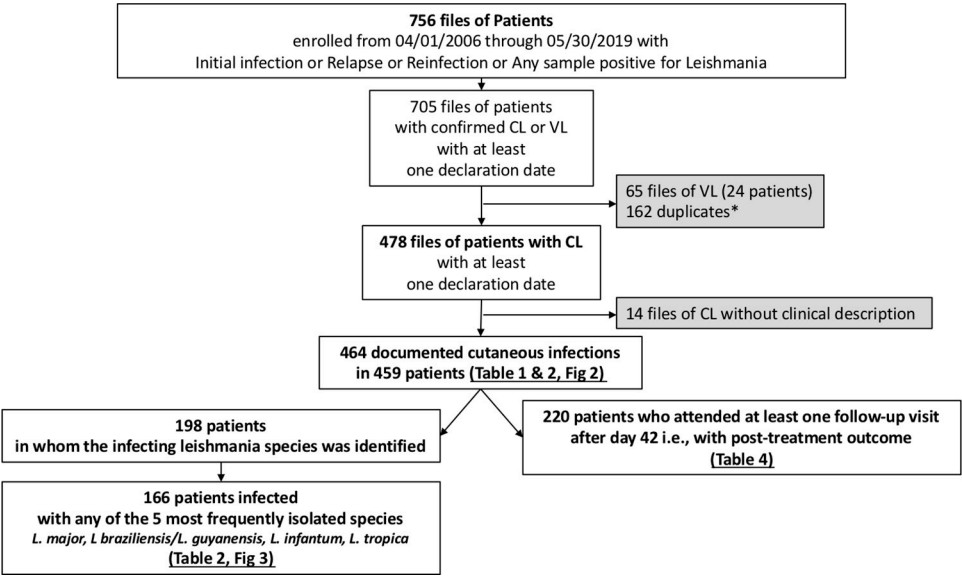

**Fig 1. Flow chart of entries in the LeishMan database and selected analyses.** Infection, Relapse or Reinfection correspond to a documented episode of visceral or cutaneous leishmaniasis, with or without mucosal involvement. * patients with multiple follow-up visits or samples. Notes: A "Patient" file corresponds to demographic information. A "Sample" file corresponds to one sample collected in one site with one technique at one date (eg. PCR on skin biopsy of right hand collected the 25th of May 2017). Abbreviations: CL, cutaneous leishmaniasis; VL, visceral leishmaniasis

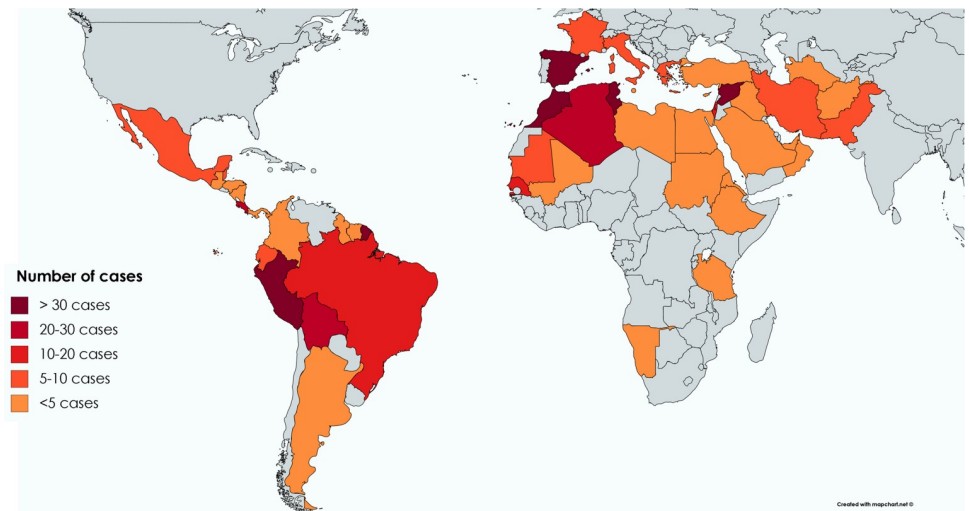

**Fig 2. Number of cases reported from each country of acquisition (464 infections in 459 patients).** Created with mapchart.net.

## Clinical features of Cutaneous Disease

Most patients had lesions limited to the skin (n = 440; 95%), with typically one or two ulcerated lesions of the limbs that had been present for 3 months prior to diagnosis and were 10 to 30 mm-wide (**Table E in S1 Table**). Compared to lesions acquired in the Old World, lesions acquired in the New World were more frequently ulcerative (75% versus 47%), larger (median diameter 30 mm versus 20 mm), more frequently localized on the limbs (61% versus 47%) and more frequently associated with nodular lymphangitis (30% versus 6%) (**Table 1**).

Identification of the infecting *Leishmania* complex in 198 patients (**Table F in S1 Table**) showed species-associated demographic and clinical patterns. For patients with no species identification, diagnosis was mostly confirmed by histology or smear, i.e., samples on which identification cannot be performed. Excluding rare species and incomplete species (**Table F in S1 Table**), we focused on the 5 most frequent infecting complex species (affecting a sub-cohort of 166 patients (**Fig 1**). This analysis showed that typically, children infected with *L. major* in Africa had rapidly evolving multiple lesions of the limbs; children and young adults infected with *L. tropica* had a lesion of the face without mucosal involvement; 40–70 year-old tourists infected with *L. infantum* in Southern Europe had a slowly evolving papulo-nodular lesion in the face and mucosal involvement in 22% of cases; young adults infected in Latin America with *L. braziliensis* or *L. guyanensis* had a rapidly evolving ulcer of the lower limbs with lymphangitis and mucosal involvement in 35% and 6% of cases, respectively (**Fig 3 and Table 2**). Immunosuppression was reported in 22 patients and was related to ongoing therapy with anti-cancer chemotherapy (n = 1), ustekinumab (n = 1), methotrexate (N = 5), TNFα antagonists (n = 5), prolonged corticosteroids with or without other immunosuppressive drugs (azathioprine (n = 1), mycophenolate mofetil (n = 1) or corticosteroids alone (n = 4)), or HIV infection (n = 3), or Good syndrome (n = 1).

## Mucosal involvement

Mucosal involvement was observed in 20 patients (4.3%), 15 were infected in the Old World and 5 in the New World (**Table 3**). In the 5 patients from the New World 2 had ML and 3 had MCL, while in the 15 patients from the Old World 7 had ML and 8 had MCL. The risk of

**Table 1. Comparative features of cutaneous leishmaniasis by continent(s) of acquisition.**

| | | New World | Old World | p-value |
|---|---|---|---|---|
| | | 185 infections | 279 infections | |
| Age, median [IQR] | | 30 [24–38] | 32 [10–58] | 0·81 |
| Male | | 75% (138) | 54% (152) | <0·01 |
| Immunocompromised | | 0·6% (1/177) | 8% (21/264) | <0·01 |
| Type of traveller | | | | NA |
| | Tourist | 61% (109/180) | 36% (96/269) | |
| | Visiting Friends and Relatives | 7% (12/180) | 43% (117/269) | |
| | Migrant | 0% (0/180) | 14% (39/269) | |
| | Expatriate (worker, missionary) | 12% (21/180) | 3% (8/269) | |
| | Soldier | 13% (24/180) | 0% (0/269) | |
| | Others | 8% (14/180) | 3% (9/269) | |
| Type of cutaneous leishmaniasis | | | | NA |
| | Localized Cutaneous | 96% (178) | 94% (262) | |
| | Muco-cutaneous | 2% (3) | 2·5% (7) | |
| | Mucosal | 1% (2) | 2·5% (7) | |
| | PKDL | 0·5% (1) | 0·3% (1) | |
| | Disseminated cutaneous leishmaniasis | 0·5% (1) | 0·3% (1) | |
| | Muco-cutaneous and visceral leishmaniasis | 0% (0) | 0·3% (1) | |
| Delay from first symptoms to the first consultation (in months), median [IQR] | | 3 [2–4] | 4 [3–7] | <0·01 |
| Number of lesions, median [IQR] | | 1 [1–2·2] | 2 [1–4] | <0·01 |
| Type of lesions | | | | < 0.01 |
| | Ulcer (wet crust) | 75·5% (139/184) | 47·5% (128/271) | |
| | Papulo-nodular | 5·5% (10/184) | 25·5% (69/271) | |
| | Dry crust | 14% (26/184) | 16% (44/271) | |
| | Squamous plaque | 3% (5/184) | 8·5% (23/271) | |
| | Others | 2% (4/184) | 1% (3/271) | |
| | Scar or new papule on a previous scar | 0% (0/184) | 1·5% (4/271) | |
| Lesion localization | | | | <0·01 |
| | Upper limb | 32% (58/183) | 31% (84/274) | |
| | Face, neck and scalp | 17% (31/183) | 38% (104/274) | |
| | Lower limb | 29% (53/183) | 16% (45/274) | |
| | Hand | 8% (15/183) | 8% (21/274) | |
| | Trunk | 7% (13/183) | 2% (6/274) | |
| | Feet | 4% (7/183) | 4% (10/274) | |
| | Neck and Scalp | 3% (6/183) | 1% (4/274) | |
| Diameter of largest lesion (millimeter), median [IQR] | | 30 [20–43·5] | 20 [10–35] | <0·01 |
| Nodular lymphangitis* | | 30% (56) | 6% (16) | <0·01 |

Notes.

*Nodular lymphangitis was defined as subcutaneous nodules in proximity to the primary lesion and/or dilated palpable lymphatic vessels in the form of a "beaded cord," and/or regional lymphadenitis

Univariate analysis. Categorical variables not included in the univariate analysis (indicated p-value as "NA"). Data are % (n) unless indicated.

Abbreviations: IQR, interquartile range; PKDL, Post Kala-azar Dermal Leishmaniasis.

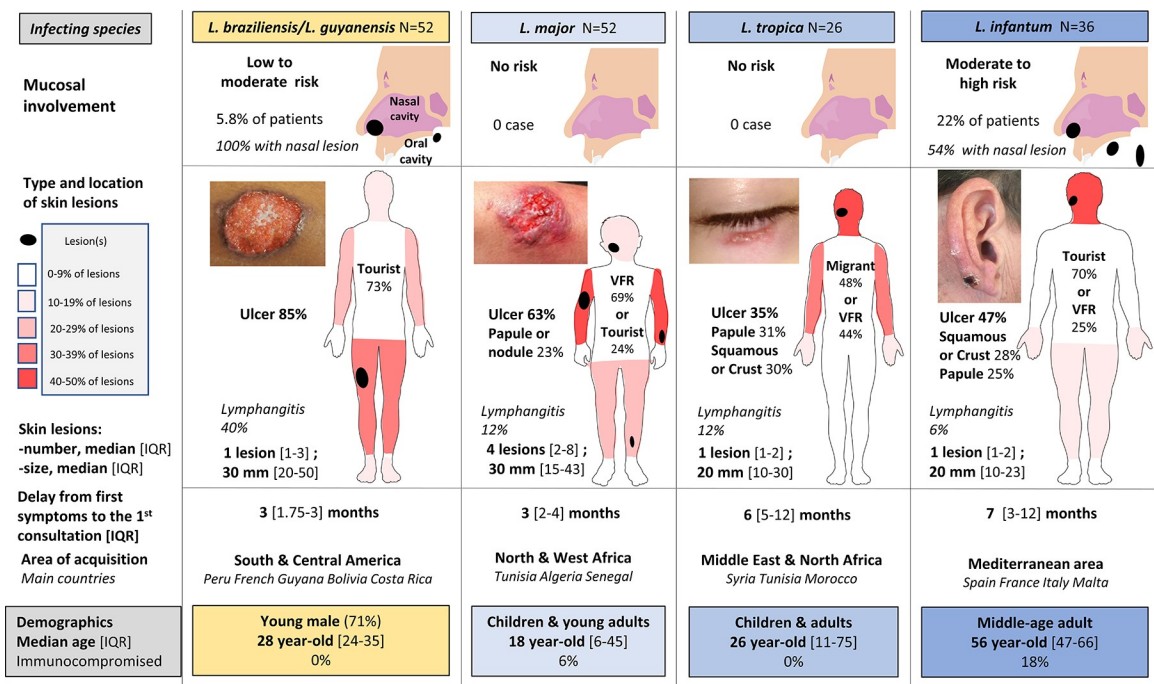

**Fig 3. Comparative features of 166 cases of cutaneous leishmaniasis by main infecting species.** Abbreviations: IQR, interquartile range; VFR, visiting friends and relatives.

having mucosal involvement at presentation was 5.3% in patients with lesions acquired in the Old World and 2.7% in patients infected in the New World. Countries of acquisition of New World mucosal or muco-cutaneous leishmaniasis were Bolivia (2 cases), Costa Rica (1 case), French Guiana (1 case), Nicaragua (1 case). For Old World leishmaniasis with mucosal involvement, countries of acquisition were Spain (7 cases), France (2 cases), Italy (2 cases), Greece (2 cases), Oman (1 case), Turkey (1 case). We observed 10 MCL, 9 ML, and one MCL with visceral involvement and inaugural skin lesions in a patient with AIDS (CD4 count 67/mm3). Mouth and laryngeal lesions were observed in 6/13 cases from Old World (missing data in 2 patients) and 1/5 case from New World, while lesions of nasal cavity were observed in 7/13 cases from Old World and 5/5 cases from New World. Seven of 15 infections (47%) with mucosal involvement in the Old World were observed in immunocompromised patients while no patient was immunocompromised in the New World subgroup with mucosal involvement. A previous history of leishmaniasis was reported in 5 of 20 patients with mucosal involvement. CL, ML or VL had been diagnosed in 1, 2 and 2 patients respectively. In all patients except one (a previous VL episode 20 years before), the mucosal involvement occurred less than 5 years (median = 4 years) after the first episode. Analysis in the subgroup of patients with an identified *Leishmania* species found a prevalence of mucosal involvement of 22%, 5.8%, 0% and 0% for *L. (infantum)/donovani* Complex, *L. braziliensis/L. guyanensis* Complex, *L. tropica* Complex and *L. major* respectively (**Fig 3** and **Table 2**).

## Local or systemic therapy

More than 10 different treatment regimens were used by physicians in the network (**Table 4**). Compared to patients who received systemic therapy, patients treated with local therapy were more frequently infected in the Old World (88% vs 40%, p< 0.0001), had smaller lesions (median 19mm vs 31 mm, p < 0.0001), less frequently associated with nodular lymphangitis

**Table 2. Comparative features of 166 cases of cutaneous, mucocutaneous and mucosal leishmaniasis by main infecting complex species.**

| | *L. braziliensis* complex n = 34 | *L. guyanensis* complex n = 18 | *L. major* complex n = 52 | *L. tropica* complex n = 26 | *L. donovani* complex n = 36 |
|---|---|---|---|---|---|
| **Age, median [IQR]** | 27 [23–33] | 32 [27–37] | 18.5 [6–46] | 26 [12–50] | 56 [47–66] |
| **Male** | 76% (26) | 61% (11) | 54% (28) | 50% (13) | 50% (18) |
| **Immunocompromised** | 0% (0) | 0% (0) | 6% (3) | 0% (0) | 18% (6) |
| **Type of traveller** | | | | | |
| **Tourist** | 73% (24) | 50% (9) | 24% (12) | 8% (2) | 70% (25) |
| **Visiting Friends & Relatives** | 3% (1) | 17% (3) | 69% (35) | 44% (11) | 25% (9) |
| **Migrants, Expatriate, Soldiers and Others** | 24 % (8) | 33% (6) | 7.8 % (4) | 48% (12) | 5.5 % (2) |
| **First 3 countries of acquisition** | | | | | |
| | Peru (12) | Costa Rica (8) | Tunisia (10) | Syria (10) | Spain (11) |
| | Bolivia (9) | French Guiana (7) | Algeria (10) | Tunisia (4) | France (4) |
| | French Guiana (5) | Brazil, Suriname, Peru (1+1+1) | Senegal (8) | Morocco (3) | Italy + Malta (4) |
| **Delay from first symptoms to the first consultation, median [IQR]** | 2 [1,5-3] | 3 [2-3] | 3 [2-3] | 6 [5-12] | 7 [3-12] |
| **Number of lesions, median [IQR]** | 1 [1-2] | 2 [1-3] | 4 [2-8] | 1 [1-2] | 1 [1-2] |
| **Type of lesions** | | | | | |
| **Ulcer (wet crust)** | 85% (28) | 88% (16) | 63% (32) | 35% (9) | 47% (15) |
| **Papulo-nodular** | 0% (0) | 6% (1) | 23% (12) | 31% (8) | 25% (8) |
| **Dry crust** | 9% (3) | 6% (1) | 12% (6) | 11% (3) | 19% (6) |
| **Squamous plaque** | 0% (0) | 0% (0) | 2% (1) | 19% (5) | 9% (3) |
| **Others** | 6% (2) | 0% (0) | 0% (0) | 4% (1) | 0% (0) |
| **Lesion localization** | | | | | |
| **Upper limb** | 18% (6) | 28% (5) | 44% (23) | 34% (9) | 24% (8) |
| **Face, neck and scalp** | 15% (5) | 16% (3) | 19% (10) | 46% (12) | 40% (13) |
| **Lower limb** | 43% (14) | 28% (5) | 29% (15) | 8% (2) | 18% (6) |
| **Hand** | 6% (2) | 22% (4) | 6% (3) | 12% (3) | 12% (4) |
| **Trunk** | 6% (2) | 6% (1) | 0% (0) | 0% (0) | 6% (2) |
| **Feet** | 12% (4) | 0% (0) | 2% (1) | 0% (0) | 0% (0) |
| **Diameter of largest lesion (mm), median [IQR]** | 30 [20-39] | 40 [25-50] | 30 [15-43] | 20 [10-30] | 7 [3-12] |
| **Nodular lymphangitis** | 35% (12) | 44% (9) | 12% (6) | 12% (3) | 6% (2) |
| **Mucosal involvement** | 6% (2) ML (1) MCL(1) | 5% (1) MCL (1) | 0% (0) | 0% (0) | 22% (8) ML(4) MCL(4) |

Note. Data are % (n) unless indicated. Abbreviations: IQR, interquartile range

7.5% vs 28.3%, p < 0.0001), and similar number of lesions (median 2 vs 1.5, p = 0.51) (**Table 4**). These different approaches applied in these different patient populations resulted in similar cure rates (89% vs 86%) at first evaluation with a median full duration of follow-up of 80 days [IQR 59–111]. As previously observed "No specific treatment", which corresponds to washing lesions with soap and water followed by semi-occlusive dressing, was associated with high (81%) cure rate [7]. Patients treated with this approach were mainly infected in OW (90%; 19/21). Due to small numbers within each subgroup, analysis of treatment outcome has not been made in respect to infecting species within the groups Old World CL and New world CL.

**Table 3. Comparative features in patients with or without mucosal involvement at presentation.**

| | No mucosal involvement | With mucosal involvement | |
|---|---|---|---|
| | 444 infections | 20 infections | p value |
| **Age, median [IQR]** | 30 [18–51] | 58 [33–65] | **<0·001** |
| **Age > 50 years** | 27% (119) | 65% (13) | **<0·001** |
| **Male** | 62% (274) | 80% (16) | 0·160 |
| **Immunocompromised** | 3% (15) | 37% (7) | **<0·001** |
| **Region of acquisition** | | | 0·250 |
| Old World | 59·5% (264) | 75% (15) | |
| New World | 40·5% (180) | 25% (5) | |
| Delay from first symptoms to the first consultation, **median [IQR]** | 3 [2–6] | 5 [3–12] | 0·057 |
| **Number of lesions, median [IQR]** | 2 [1–3] | 1.5 [1–3] | 0·287 |
| **Face, neck or scalp involvement (excluding pure mucosal forms)** | 29% (123/428) | 64% (7/11) | **0.020** |
| **Large lesion**<sup></sup>* | 50% (196) | 64% (7) | 0.560 |
| **Nodular lymphangitis** | 15·5% (69) | 15% (3) | 1 |

Data are % (n) unless indicated.

*: > 4 cm2 (corresponding to a diameter of 23mm)

Abbreviations: IQR, interquartile range.

**Table 4. Clinical characteristics of patients and healing rates according to treatment administration modality (systemic versus local).**

| | Local treatment | | Systemic treatment | | |
|---|---|---|---|---|---|
| | n = 107 | | n = 113 | | p value |
| **Patients infected in Old World** | 94 (88) | | 45 (40) | | **<0,0001** |
| **Age, median [IQR]** | 30 [15–57] | | 30 [21–51] | | 0·71 |
| **Number of lesions, median [IQR]** | 2 [1–4] | | 1·5 [1–3] | | 0·51 |
| **Diameter of largest lesion** (mm), **median [IQR]** | 19 [10–30] | | 31 [20–50] | | **<0·0001** |
| Delay from first symptoms to the first consultation, **median [IQR]** | 4 [3–6] | | 3 [2–4] | | **0·008** |
| **Nodular lymphangitis** | 8 (4,5) | | 32 (28,5) | | **<0·0001** |
| **Immunosuppression or diabete mellitus** | 12 (11) | | 5 (4,5) | | 0·05 |
| **Healing rate** | 95/107 (89) | | 97/113 (86) | | 0·05 |
| | **Treatment received** | **Healing rate** | **Treatment received** | **Healing rate** | |
| | No specific treatment * | 17/21 (81) | Antimonial therapy (MA or SSG) | 30/34 (88) | |
| | Intralesional MA or SSG +/- Cryotherapy ** | 70/75 (93) | Amphotericin B (liposomal or deoxycholate***) | 7/12 (58) | |
| | Topical Paromomycin | 7/9 (78) | Fluconazole | 11/12 (92) | |
| | Others: surgery (n = 1), imiquimod (n = 1) | 1/2 (50) | Miltefosine | 38/43 (88) | |
| | | | Pentamidine Isethionate | 10/11 (91) | |
| | | | Other: SSG + pentoxifylline (n = 1) | 1/1 (100) | |

Data are n (%) unless indicated. Healing rate = first evaluation after at least 42 days of treatment initiation (median follow-up of 80 days [IQR 59–111]).

Note

*Wash lesion and wound dressing

**two patients received cryotherapy alone

*** all patients except one received liposomal Amphotericin B

Abbreviations: MA, meglumine antimoniate; SSG, sodium stibogluconate

### Patients treated with non-systemic therapy

In patients with CL acquired in the New World, healing after day 42 occurred in 12 of 13 (92%) managed with local therapy, and in 55 of 68 (81%) managed with systemic therapy (p = 0.32). In patients with Old World CL, healing after day 42 occurred in 83 of 94 (88%) managed with local therapy and in 42 of 45 (93%) managed with systemic therapy.

### Patients with CL eligible for local therapy

Among 348/440 patients with localized CL and all available clinical criteria (see Methods), we found that at least 208/348 patients (60%) were eligible for local therapy. Among them, 90 and 118 patients had been infected in the New World and Old World respectively.

## Discussion

Cutaneous Leishmaniasis (CL) displays a diversity of lesion aspect, number, size, and location on the body surface, including the potential extension to mucosae of the nose, mouth, pharynx and larynx. These features determine the selection of the most appropriate treatment regimens, from simple wound dressing to long courses of potentially toxic parenteral drugs. In this cohort of 459 European patients with CL infected in 47 countries from 4 continents, we show that mucosal involvement (that includes both purely mucosal or mixed cutaneous and mucosal infections) was present in less than 5% of patients and was similarly infrequent in subjects infected either in Latin America ("New World", 2.7%) or in Europe, Africa or Asia ("Old World", 5.3%). Forms caused by the 5 most frequent infecting complex species (*L. major*, *L. tropica*, *L. infantum*, *L. braziliensis*, and *L. guyanensis*), identified in 166 patients, were each associated with distinct features. In particular, no mucosal involvement was observed in patients infected with either *L. major* or *L. tropica*, whereas it affected 6% of patients infected with *L. braziliensis* or *L. guyanensis complex* (2 MCL, 1 ML) and 22% of patients infected with *L. infantum* (4 ML, 4 MCL), in whom this complication was strongly (though not exclusively) related to pre-existing immunosuppression. This new set of data on lesions characteristics and on the risk of mucosal involvement will form a solid basis to refine treatment recommendations, based on an optimized benefit-to-risk analysis.

We observed 4 typical patterns of CL presentation. Young adults infected in the New World with *L. braziliensis* or *L. guyanensis* complex typically had a single, large ulcerative lesion on the lower limb. *L. major*- and *L. tropica*-infected subjects were predominantly children and young adults, who typically had either multiple lesions located on the upper limbs or a single lesion on the face. *L. infantum*-infected subjects were predominantly older than 40 years, and typically had a single small lesion in the face. Interestingly, travelers returning from South and Central America are predominantly young backpackers visiting deep forests. In terms of age and lesion characteristics, this population resembles that of young infected native patients living in areas where leishmaniasis is endemic. Taken together, these clinical patterns should help attending physicians recognize typical forms of CL, hopefully reducing the diagnostic delay that currently averages 3 months. Discriminating the respective contributions of parasite-, vector- and host-related factors in these new phenotypes is beyond the scope of our current approach but the sustained, multisite, multiparametric surveys performed by the LeishMan network and other groups worldwide will provide information to more precisely revisit the pathogenesis of human leishmaniasis [9].

It was surprising that infections acquired in the Old World (and more specifically in Europe or some Mediterranean countries) had higher rates of mucosal involvement than infections with *L. braziliensis* or *L. guyanensis* complex acquired in Latin America. The prevalence of mucosal involvement at baseline was 4.3% in the cohort which is consistent with the rate

(usually <7%) observed in studies including travellers [5,16]. In our study, the unexpected, low rate of mucosal involvement in infections acquired in the New World contrasts with the unexpected, relatively high rate in infections acquired in the Old World. High rates of mucosal involvement patients with Old World leishmaniasis, especially in *L. infantum* infection, has been recently observed in some but not all studies in travellers [17–19]. The increasing number of immunocompromised patients, probably more keen to travel to European countries than to Latin America, likely contributes to the rising incidence of this complication [20]. A previous study from France also suggested a risk of visceral dissemination that we did not observe in our large cohort [21]. Immunosuppression seems to play an important role in the pathogenesis of ML in the Old World, while its role in ML in New World is more difficult to determine, as many apparently immunocompetent patients do develop ML.

Systemic treatment is warranted in patients with *L. infantum* mucosal infection, in whom miltefosine and liposomal amphotericin B seem effective [8,22]. Therefore, in immunosuppressed patients with cutaneous lesions due to *L. infantum*, miltefosine and liposomal amphotericin B should probably be used, in the hope of preventing the subsequent occurrence of mucosal involvement that is frequently multifocal with this species (Fig 3). Of interest however, we confirm here recent observations showing that liposomal amphotericin B is not highly effective in CL caused by *Leishmania* species other than *L. infantum* [8,23].

In the subpopulation of patients in whom it was used, local therapy was effective. Compared to patients who received systemic therapy, patients treated with local therapy were more frequently infected in the Old World, had smaller lesions, and less frequent nodular lymphangitis. Arguably, these differences reflect predominantly treatment choices by physicians, based on the continent of infection and, to a lesser extent, on the applicability of local treatment. Local and systemic therapy were thus used in different patient populations but resulted in similarly high cure rates (89% vs 86%). This tells little on the intrinsic power of these different approaches but suggests that local treatment was appropriate in most patients in whom it was used, i.e., that criteria defined in the LeishMan consensus were accurate [11,12,16]. Local therapy is now proposed by national and international recommendations not only for Old World but also for a proportion of patients with New World CL [24]. In the small number of patients with New World CL treated locally, healing was almost the rule in this cohort, suggesting that more patients may benefit from this approach in the future. These results match those of prospective studies in endemic countries but larger prospective studies are needed before a robust conclusion can be reached in the specific context of CL in travellers [16,25–27]. Not all patients may be eligible to local therapy. In particular, whether infection acquired in Bolivia is compatible with local therapy is still a matter of controversy. A higher risk of mucosal involvement in travellers returning from Andean countries especially Bolivia has indeed been reported [16].

Solomon and colleagues found that 17 of 145 (11.7%) Israeli travellers with CL acquired in Amazon Basin in Bolivia received a diagnosis of ML [28]. We observed a prevalence of 9.5% (2/21) of mucosal involvement in travellers from Bolivia in our study, which matches the 5–15% rate indicated in a quasi-exhaustive review of the literature and the 11.5% rate reported in a recent large cohort of travellers by Boggild and al. [5,16]. Detecting subsequent mucosal relapse through an extended follow-up was beyond the immediate scope of our study. However, as most teams involved in the LeishMan consortium are reference centres for leishmaniasis in their respective country, they are expected to attend most patients with secondary mucosal involvement, and capture corresponding information in the common database. The similar risks of mucosal involvement across these different reports suggest that very few if any cases of were missed by our consortium during the study period. In our cohort, a previous episode of leishmaniasis was reported in 20% of patients with mucosal involvement. Several teams in our consortium give to CL patients a note mentioning the diagnosis of leishmaniasis

and the significant (though relatively low) risk of oro-nasal complications that may occur years to decades after the initial episode. Our results suggest that a similar information should also be provided after a VL episode.

Approximately two thirds of patients of our cohort were eligible for local therapy when following guidelines in travellers based on lesions number, size, location and preexisting conditions [7,12,15]. Even though there may have been some heterogeneity in therapeutic management during the study period, the general algorithm we described in 2013 has remained our consortium's therapeutic management backbone. Cryotherapy followed by intralesional injections of pentavalent antimony is difficult to perform on several skin locations (lips, eyelids, genitalia, hands and feet), and in remote endemic areas where liquid nitrogen is not available and where injections are potentially harmful. In young patients with many and/or large lesions this effective yet painful approach is vastly suboptimal [11]. By contrast, application of paromomycin-based ointments is limited only by the site of application (ie. peribuccal and eyelids), and is well accepted by children and patients with many lesions. It has proved effective in several randomized trials in the Old and New World [25,26,29,30]. Gaps and issues in its development have been thwarted by some teams who developed local yet effective paromomycin formulations, including in Bolivia (Guéry & Buffet, personal data) [27].

This study has limitations. The number of mucosal or mucocutaneous episodes was too small to enable a robust multivariate analysis of the risk factors for mucosal involvement. Because this is a common surveillance program and not a prospective, comparative clinical trial, we could not capture follow-up data in all patients, hence the need for a careful interpretation of our observations regarding therapy. Not least, HIV testing was not systematically done which may limit our conclusions regarding the impact of immunosuppression on the clinical spectrum of cutaneous leishmaniasis. Nevertheless, this robust description of clinical, parasitological and therapeutic features of cutaneous leishmaniasis in almost half a thousand of travelers, reveals a moving landscape, where the risk of mucosal involvement is not limited to travels in the New World and effective treatments of CL are not limited to systemic therapy.

## Supporting information

**S1 Table. (A). Number of patients attended at each centre of the LeishMan consortium. (B). Number of cases according each country of LeishMan consortium.** Abbreviations: UK, United Kingdom. **(C). Suspected countries of acquisition in Old World. (D). Suspected countries of acquisition in New World. (E). Demographic and clinical characteristics of total cohort.** Note. Results are expressed as number (%), unless otherwise stated. There are some missing data for each variable (<10%) explaining incomplete count proportion for categorical variables. Abbreviations: DissCL, disseminated cutaneous leishmaniasis; IQR, interquartile range; MCL, muco-cutaneous leishmaniasis, PKDL, post Kala-azar dermal leishmaniasis; VL, visceral leishmaniasis. **(F). Subgenus, complex species, species identified in 198 cutaneous leishmaniasis infections.** Note. *Viannia subgenus not further identified to species level.
(DOC)

**S1 Fig. Age distribution of the main categories of travellers.** Panel A: age of distribution in total cohort; Panel B: age distribution in travellers visiting friends and relatives; Panel C: age distribution in tourists. Abbreviations: VFR visiting friends and relatives.
(TIF)

**S1 Data. Dataset 1.**
(CSV)

**S2 Data. Dataset 2.**
(CSV)

## Acknowledgments

We thank Sandra Manceau and Gloria Morizot for helpful assistance. We thank members of the Leishman Network and the French Cutaneous Leishmaniasis Study group who contributed to the study:

| First Name | First Initial | Surname |
|---|---|---|
| Isabelle | I | ALCARAZ |
| Florence | F | AMELOT |
| Fanny | F | ANDRY |
| Adela | A | ANGOULVANT |
| Julie | J | ARATA BARDET |
| Yercanik | Y | ARMAGAN |
| Jean Philippe | JP | ARNAULT |
| Selma | S | AZIB-MEFTAH |
| Dominique | D | BARTHELME |
| Bernhard | B | BECK |
| Sorya | S | BELAZ |
| Muriel | M | BELLIAH-NAPPEZ |
| Nathalia | N | BELLON |
| Nathalie | N | BENETON |
| Astrid | A | BLOM |
| Olivia | O | BOCCARA |
| Nathalie | N | BODAK |
| Mathilde | M | BON MARDION |
| Daphné | D | BOSSET |
| Emmanuel | E | BOTTIEAU |
| François | F | BOUQUEAU |
| Emmanuelle | E | BOURRAT |
| Romain | R | BRICCA |
| Olivier | O | CARPENTIER |
| Eline | E | CASASSA |
| Eric | E | CAUMES |
| Sophie | S | CHARLES |
| Angèle | A | CLABAUT |
| Heloise | H | CLERC |
| Hélène | H | COIGNARD |
| Domenica | D | CUSTER |
| Hervé | H | DARIE |
| Madeleine | M | DE DARUVAR |
| Henry JC | HJC | DE VRIES |
| Sebastien | S | DEBARBIEUX |
| Marine | M | DELOBEAU |
| Charlotte | C | DENTAN |
| Julie | J | DI LUCCA |
| François | F | DURUPT |

(*Continued*)

| | | |
|---|---|---|
| Olivier | O | EPAULARD |
| Catherine | C | ESCHARD |
| Jean-Louis | JL | ESTIVAL |
| William | W | ETIENNE |
| Marie | M | FERNEINY |
| Fanny | F | FICHEL |
| Françoise | F | FOULET |
| Alexandre | A | GALLOULA |
| Florence | F | ROBERT-GANGNEUX |
| Amélie | A | GANTZER |
| Gilles | G | GARGALA |
| Pauline | P | GELOT |
| Céline | C | GIRARD |
| Hedvig | H | GLANS |
| Jeremy | J | GOTTLIEB |
| Emmanuel | E | HEAU |
| Michel | M | JANIER |
| Kaoutar | K | JIDAR |
| Judith | J | KARSENTY |
| Christine | C | KATLAMA |
| Natalia | N | KIRSTEN |
| Diane | D | KOTTLER |
| Nora | N | KRAMKIMEL |
| Barthélémy | B | LAFON-DESMURS |
| Brigitte | B | LAGRANGE |
| Pauline | P | LANSALOT-MATRAS |
| Noël | N | LE CAM |
| Laurence | L | LE CLEACH |
| David | D | LEBEAUX |
| Delphine | D | LEBRUN |
| Maeva | M | LEFEBVRE |
| Regine | R | LEVET |
| Audrey | A | LORRIAUX |
| Devy | D | LU |
| Hervé | H | MAILLARD |
| Mylene | M | MAILLET |
| Fredrik | F | MANSSON |
| Guillaume | G | MARTIN-BLONDEL |
| Valérie | V | MARTINEZ-POURCHER |
| Pierre | P | MARTY |
| Guillaume | G | MELLON |
| Oussama | O | MOURI |
| Justine | J | MUNOZ |
| Kristina | K | OPLETALOVA |
| André | A | PAUGAM |
| Alice | A | PERIGNON |
| Jean-Luc | JL | PERROT |
| Eskild | E | PETERSEN |

| | | |
|---|---|---|
| Antoine | A | PETIT |
| Catherine | C | PICARD-DAHAN |
| Emily | E | POLLOCK |
| Christelle | C | POMARES |
| Charlotte | C | POUPLARD |
| Olivier | O | ROGEAUX |
| Mahtab | M | SAMIMI |
| Mathilda | M | SANDSTROM |
| Anne | A | SAUSSINE |
| Pierre | P | SCHNEIDER |
| Robert | R | SEBBAG |
| Patricia | P | SENET |
| Amandine | A | SERVY |
| Patrick | P | SOENTJENS |
| Cornelis | C | STIJNIS |
| Marc | M | THELLIER |
| Feyrouz | F | TOUKAL |
| Mathis | M | TREPP |
| Raoul | R | TRILLER |
| François | F | TRUCHETET |
| Aude | A | VALOIS |
| Steven | S | VAN DEN BROUCKE |
| Jef | J | VAN DEN ENDE |
| Erwin | E | VAN DEN ENDEN |
| Alfons | A | VAN GOMPEL |
| Pieter-Paul | PP | VAN THIEL |
| Michèle | M | VAN VUGT |
| Francisco | F | VEGA-LOPEZ |
| Filip | F | VERHAEGHE |
| Mireille | M | VERNIER |
| Michele | M | WOLTER-DESFOSSES |
| Daniela | D | ZAHARIA |
| Stéphane | S | ZEMIRO |
| Elina | E | ZUELGARAY |

## Author Contributions

**Conceptualization:** Romain Guery, Gundel Harms, Johannes Blum, Diana N. Lockwood, Pierre Buffet.

**Data curation:** Romain Guery, Gert Van der Auwera, Pierre Buffet.

**Formal analysis:** Romain Guery, Stephen L. Walker, Gundel Harms, Andreas Neumayr, Pieter Van Thiel, Jean-Pierre Gangneux, Jan Clerinx, Sara Karlsson Söbirk, Leo Visser, Laurence Lachaud, Mark Bailey, Aldert Bart, Christophe Ravel, Johannes Blum, Pierre Buffet.

**Methodology:** Romain Guery, Pierre Buffet.

**Project administration:** Gert Van der Auwera, Johannes Blum, Pierre Buffet.

**Supervision:** Johannes Blum, Diana N. Lockwood, Pierre Buffet.

**Validation:** Romain Guery, Stephen L. Walker, Gundel Harms, Andreas Neumayr, Pieter Van Thiel, Jean-Pierre Gangneux, Jan Clerinx, Sara Karlsson Söbirk, Leo Visser, Laurence Lachaud, Mark Bailey, Aldert Bart, Christophe Ravel, Gert Van der Auwera, Johannes Blum, Diana N. Lockwood, Pierre Buffet.

**Writing – original draft:** Romain Guery, Pierre Buffet.

**Writing – review & editing:** Stephen L. Walker, Gundel Harms, Andreas Neumayr, Pieter Van Thiel, Jean-Pierre Gangneux, Jan Clerinx, Sara Karlsson Söbirk, Leo Visser, Laurence Lachaud, Mark Bailey, Aldert Bart, Christophe Ravel, Gert Van der Auwera, Johannes Blum, Diana N. Lockwood, Pierre Buffet.

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
