## [Decision Letter · Decision Letter 0]

22 Jan 2021

Dear Mr. Guery,

Thank you very much for submitting your manuscript "Clinical diversity, mucosal involvement and treatment strategy in cutaneous leishmaniasis: a European clinical report in 459 patients." for consideration at PLOS Neglected Tropical Diseases. As with all papers reviewed by the journal, your manuscript was reviewed by members of the editorial board and by several independent reviewers. In light of the reviews (below this email), we would like to invite the resubmission of a significantly-revised version that takes into account the reviewers' comments. 

We cannot make any decision about publication until we have seen the revised manuscript and your response to the reviewers' comments. Your revised manuscript is also likely to be sent to reviewers for further evaluation.

Sincerely,

Hechmi Louzir, M.D

Associate Editor

Edgar Carvalho

Deputy Editor

Reviewer's Responses to Questions

**Key Review Criteria Required for Acceptance?**

**Methods**

-Are the objectives of the study clearly articulated with a clear testable hypothesis stated?

-Is the study design appropriate to address the stated objectives?

-Is the population clearly described and appropriate for the hypothesis being tested?

-Is the sample size sufficient to ensure adequate power to address the hypothesis being tested?

-Were correct statistical analysis used to support conclusions?

-Are there concerns about ethical or regulatory requirements being met?

Reviewer #1: -Are the objectives of the study clearly articulated [YES] with a clear testable hypothesis stated[NOT APPLICABLE]?

-Is the study design appropriate to address the stated objectives [YES]?

-Is the population clearly described and appropriate for the hypothesis being tested[YES]?

-Is the sample size sufficient to ensure adequate power to address the hypothesis being tested [NOT APPLICABLE?

-Were correct statistical analysis used to support conclusions[YES]?

-Are there concerns about ethical or regulatory requirements being met[NO]?

Reviewer #2: This is an elegant, well-conceived, well-planned, and well-executed work. Provided information is very useful in the understanding of relationships and differences between the Old and New World integumentary forms of leishmaniasis and includes a very good number of patients with complete information, which is usually a major deficiency in case series reports. No observation or concerns regarding ethics and statistic analysis. Processes and steps are clearly mentioned, definitions and outcomes are precise.

Reviewer #3: The objectives are clear and the design study is appropriated for the hypotesis. It is a convenience cohort study, with 459 patients included, however the number of patients approached is not described in the methodoly, only in teh supplement file. Please make this information in teh text.

Reviewer #4: This is an interesting report on clinical and therapeutic aspects of travelers acquiring cutaneous leishmaniasis in both New and Old World countries. Data were systematically collected by members of a consortium following a harmonized protocol during two decades. The objectives are clear and the study was mainly descriptive with some comparisons between Old and New World acquired cases and also between cases due to different parasite species. 

In spite of the clear usefulness of the presented data for improving clinical care of such group of patients, some issues should be properly addressed in order to make clear for the potential readers the limitations of the exploratory analysis performed for possible determinants of the mucosal involvement and the lack of data on parasite species and treatment approach of a relevant number of the included patients.

Then, I strongly suggest the following: 

1. The inclusion of the details of the multivariate approach for exploring the possible ML determinants in the method section, e.g. the criteria for inclusion of the variables in the final model (it is unusual just include de variables with p<0.05 significance level in the univariate analysis without other considerations related to the biological plausibility), the exploration of confusion and potential interactions between variables and the final evaluation of the model fitness.

2. A description of the loss of data for at least the therapeutic approach and parasite identification in the flow chart (figure 1), i.e., how many patients were analyzed in accordance of parasite species? and how many patients were included for the therapeutic approach comparison?

**Results**

-Does the analysis presented match the analysis plan?

-Are the results clearly and completely presented?

-Are the figures (Tables, Images) of sufficient quality for clarity?

Reviewer #1: Does the analysis presented match the analysis plan?[THESE STANDARD PLOS QUERIES ABOUT STATISTICS ARE NOT APPLICABLE FOR THIS DESCRIPTIVE STUDY]

-Are the results clearly and completely presented[YES]?

-Are the figures (Tables, Images) of sufficient quality for clarity [YES]?

Reviewer #2: 111: "...a previous history of leishmaniasis was reported by 8% of patients in the cohort". These were patients of CL with a previous episode of CL or ML patients with a previous episode of CL? Or ML with a previous episode of ML?

114, Table 1: Demographic and clinical characteristic in this population is strikingly similar to that of infected native patients probably because travellers coming to latinamerican countries are young people looking for extreme experiences and, for such reason, are going to deep forests where this zoonotic disease is endemic and where most of native patients are infected.

114, Table 1: The low number of patients who traveled to the New World and developed mucosal lesions could be explained by the fact that several years are usually necessary between skin lesions and mucosal compromise (range 1 to 50 or more but usually 2 to 5). Do you agree or do you have another explanation? Many of your patients went to countries like Peru and Bolivia with high prevalence of mucosal disease and your results might suggest that mucous extension is not so frequent. 

114, Table 1: "Delay from first symptoms to parasitological diagnosis (months)". Why this 2 or 4 months delay? Aren't patients looking for diagnosis? Or tests to confirm diagnosis are not easily available? It could be interesting to know which other diagnostics received before confirmation of leishmaniasis, specially because such info could be helpful for doctor with not wide experience with leishmaniasis, to keep in mind all possible differential diagnosis.

114, Table 1: Includes SCAR as a type of lesion could be confussing because scar is usually the final result of leish when cured. Please explain me why was included and what happen with those 4 out of 271 OW subjects reported.

135: "lymphangitis 35%" is really high. Is this "pure" leish lymphangitis or bacterial co-infections that are quite common and also produce lymphangitis?

137, Table 2: "Duration of disease". Is this from first symptom to the consultation? Do you have info about time from the trip and first symptom? It could be useful for other doctors to know this "incubation" time to make the differential diagnosis exercise.

161: In native patients age is also a very correlated factor in mucosal disease, although most of them had initial infection (with or without cutaneous lesion) when were younger. In your over 50 patients ML was close in time to CL? In natives ML usually need a long time to appaer, what in yours?

183, Table 4: 107 patients treated with local therapies and 113 with systemic. What happen with the other 239 patients?

183, Table 4: "No specific treatment". Is this spontaneous cure? Because soap, water and dressing are good for any ulcer but 81% of cures deserve an explanation.

Reviewer #3: The analyzes presented are in accordance with the proposed methodology and the results are clearly presented.

Regarding the results some points need to be clear. 

1) In the table 1, the authors shown the following results: Delay from first symptoms to parasitological diagnosis (months), median [IQR] [1-3] 4 [2-6] <0·01. Duration of disease (in months), median [IQR] 3 [2-4] 4 [3-7] <0·01. How to explain that the delay in diagnosis is longer than the time of illness?

2) The authors mention that ulcerated injuries were more frequent in travelers traveling from the new world (75.5%) than from the old world (47.5%), but there was no statistical difference. How to affirm or explain this finding?

3) Likewise, they state that the lesions are more frequently localized on the limbs (61% versus 47%) and

more frequently associated with nodular lymphangitis (30% versus 6%), comparing New and Old world. there is no statistical difference presented.Please, clarify these point.

4) Another important point is relaed to treatment. The most interesting data would be to evaluate the therapeutic response in relation to the species of Leishmania. I believe that you cannot compare systemic treatment with local, if there is no well-defined parameter. For example: compare response to different treatments in patients with similar lesions.

Reviewer #4: 1. The most curious result was related to the mucosal involvement rate detected in patients infected with L. infantum. Also, the evidence of immunocompromise as a factor associated with such an involvement. However, the merit of the comparison of the L. infantum group against the New World acquired cases infected with parasites belonging to L. braziliensis complex should be revised. In fact, epidemiology and pathogenesis of ML associated to L. braziliensis complex is pretty different of the pattern described for the L. infantum mucosal disease. The relevant comment on that is slightly mentioned in the discussion when authors refer to the long-term follow-up needed for concluding on ML rates for patients who received local treatment. It could be misleading if readers were induced to consider that ML is more relevant for L. infantum infected patients. Instead of that, the results of this study confirm that for immunocompetent individuals ML continues to be more relevant in cases infected with parasites belonging to L. braziliensis complex. Then, an additional analysis should be done excluding the immunocompromised patients.

2. The results of the multivariate analysis for ML determinants should be summarized in a table with the unadjusted and adjusted ORs and their respective 95%CI.

3. Therapeutic data were available from 220 patients. This represents a large data loss of ~50%. An explanation should be offered and this issue should be highlighted as a relevant study limitation. 

4. Were all the patients tested for HIV infection? If not all the patients had an HIV test done, it would be mention also among the limitations of the study.

5. What type of immunocompromise were registered? This is a relevant information that could appear at least as a supplementary file.

6. Finally, the standard of care could be changed during the study period and this aspect deserves at least a comment in the discussion section.

**Conclusions**

-Are the conclusions supported by the data presented?

-Are the limitations of analysis clearly described?

-Do the authors discuss how these data can be helpful to advance our understanding of the topic under study?

-Is public health relevance addressed?

Reviewer #1: -Are the conclusions supported by the data presented [YES]?

-Are the limitations of analysis clearly described [YES]?

-Do the authors discuss how these data can be helpful to advance our understanding of the topic under study [YES]?

-Is public health relevance addressed [YES]?

Reviewer #2: 213: I agree with you that clinical characteristics are changing and that therapeutic recomendations must be reviewed in the near future. However a new group of patients from L.b regions, mainly Brazil, Perú and Bolivia, with severe, extensive, chronic and recurrent disease are emerging. According with your data (and I agree) around 75% of CL could be treated with local, simple, safe and cheap therapies. And systemic drugs have to be reserved to treat other forms of CL, ML, recurrent or relapsing diseases, leish in immunocompromised hosts, etc.

223: Yes, this comment goes in the same line with my previous (line 114) and that's why a list of initial diagnosis done before to confirm leish could be very useful for doctors with less experience in this disease.

271: "Detecting subsequent mucosal relapse through an extended follow-up was beyond the immediate scope of our study". OK, but at least you could tell us if any of the patients attended during the initial years developed ML during the time when the study was still active. My concern is because people with no big experience in CL and ML can read the paper and understand that ML appears soon after CL and forget long-term surveillance.

Reviewer #3: The main conclusion of this manuscritp is be immunocompromised and be over fifty years old are risk factors to develop mucosal leishmaniasis. 

The authors not mentioned the limitations of manuscript. 

The topic addressed in this manuscript is very important for public health, mainly to alert health professionals from non-endemic areas to leishmaniasis.

Reviewer #4: Most of the conclusions are supported by the data. The study limitations should be clearly stated as requested in above the comments . After considering the study limitations, the abstract section deserves some attention to be less emphatic for conclusions based on limited data.

**Editorial and Data Presentation Modifications?**

Reviewer #1: The Title “Clinical diversity, mucosal involvement and treatment strategy in cutaneous leishmaniasis: a European clinical report in 459 patients” is a little off. I think the authors mean “Clinical diversity, mucosal involvement and treatment RESULTS in cutaneous leishmaniasis: a European clinical report OF 459 patients.” Note that the title is not really accurate with respect to mucosal involvement, because it implies that all cases of mucosal involvement were seen in concomitant cutaneous disease, but I am not sure how to fix this. “Clinical diversity, mucosal involvement and treatment RESULTS in cutaneous and mucosal leishmaniasis” is redundant (of course there is mucosal involvement in mucosal leishmaniasis). “Clinical diversity and treatment RESULTS in cutaneous, mucocutaneous, and mucosal leishmaniasis:” is probably best.

The heading “Clinical Features” at line 122 should be “Clinical Features of Cutaneous Disease.

“ Mucosal Involvement”, the second focus of the report, is tricky. Mucocutaneous disease (mucosal disease with cutaneous disease) and mucosal disease per se (only mucosal disease) are correctly defined in Definitions, but it might be there added that mucosal involvement includes both presentations. I found it hard to track mucocutaneous disease separately from mucosal disease. In Table 2, “mucosal disease” might be replaced by “mucocutaneous disease” to make it clear that for Table 2 (cutaneous disease), all “mucosal disease” was mucocutaneous. In the paragraph beginning on line 145, the sentence “We observed 10 MCL, 9 ML, and one MCL with visceral involvement and inaugural skin lesions in a patient with AIDS (CD4 count 67/mm3)” is welcome because it separates the several presentations. It would be helpful if in the following sentences “Mouth and laryngeal lesions were observed in 6/13 cases from Old World and 1/5 case from New World, while lesions of nasal cavity were observed in 7/13 cases from Old World and 5/5 cases from New World. Seven of 15 infections (47%) with mucosal involvement in the Old World were observed in immunocompromised patients while no patient was immunocompromised in the New World subgroup with mucosal involvement”, the authors could give an idea of the breakout of mucocutaneous vs mucosal disease. In Discussion, it is said that “mucosal involvement was observed in…22% of patients infected with L. infantum, in whom this complication was strongly related to pre-existing immunosuppression.” At least this reviewer could not figure out the L infantum cases. Were these mucosally-involved cases mucosal spread from a cutaneous focus (mucocutaneous disease) or frank mucosal disease without other organs (skin, viscera) being involved? In spite of these issues, the sentence in Discussion “In our study, the unexpected, low rate of mucosal involvement in infections acquired in the New World contrasts with the unexpected, relatively high rate in infections acquired in the Old World” is both cleverly worded and accurate.

Reviewer #2: Not needed

Reviewer #3: The authors mentioned that 198 species of Leishmania were identified, but in the results and discussion, there are references to only 166 isolates. Please correct or explain this difference.

One conclusion was missing from this manuscript, electing the main findings that contribute to current literature.

Reviewer #4: (No Response)

**Summary and General Comments**

Reviewer #1: PNTD-D-20-01699

Guery et al report on “Demographic and clinical data from 459 travellers infected in 47 different

countries were collected by members of the European LeishMan consortium.” [Abstract]. 

The authors are right that “[This] harmonized data collection by clinicians attending patients infected in many transmission foci worldwide enables direct comparisons of clinical patterns induced by different Leishmania species, and the outcome following treatment with either local or systemic regimens”[Intro] and the implication, that this comparison within one report will be frequently quoted in preference to separate reports on each Leishmania species, is also right. A second focus is on mucosal disease. 

This complex report is extremely well written, especially since the first and senior authors are not native-English speakers. Lines 129-136 summarize the massive amount of demographic and clinical presentation data for cutaneous disease (tables 1-2 and figures 1-3) nicely. The presentation of mucosal disease is in Table 3 and the results of therapy for all presentations are in Table 4.

Discussion is thoughtful.

Reviewer #2: Congratulations for an excellent work; I enjoy and learn reading it.

Reviewer #3: I added my comments and sugestion for the authors, mainly in the section of results.

Reviewer #4: The study is relevant, the information is important for clinicians dealing with travelers with cutaneous lesions suspected of having leishmaniasis. It seems to me that this is the report of the largest clinical cohort of travelers with leishmaniasis ever reported in the literature.

PLOS authors have the option to publish the peer review history of their article (what does this mean?). If published, this will include your full peer review and any attached files.

Reviewer #1: Yes: Dr Jonathan D Berman

Reviewer #2: Yes: Jaime Soto, MD. FUNDERMA - Fundación Nacional de Dermatología, Santa Cruz de la Sierra, Bolivia. jaime.soto@infoleis,com

Reviewer #3: No

Reviewer #4: Yes: Gustavo Romero
---

## [Decision Letter · Decision Letter 1]

10 Aug 2021

Dear Mr. Guery,

Thank you very much for submitting your manuscript "“Clinical Diversity and Treatment Results in Cutaneous, Mucocutaneous, and Mucosal Leishmaniasis: a European clinical report in 459 patients." for consideration at PLOS Neglected Tropical Diseases. As with all papers reviewed by the journal, your manuscript was reviewed by members of the editorial board and by several independent reviewers. The reviewers appreciated the attention to an important topic. Based on the reviews, we are likely to accept this manuscript for publication, providing that you modify the manuscript according to the review recommendations. 

Sincerely,

Edgar M Carvalho

Deputy Editor

Reviewer's Responses to Questions

**Key Review Criteria Required for Acceptance?**

**Methods**

-Are the objectives of the study clearly articulated with a clear testable hypothesis stated?

-Is the study design appropriate to address the stated objectives?

-Is the population clearly described and appropriate for the hypothesis being tested?

-Is the sample size sufficient to ensure adequate power to address the hypothesis being tested?

-Were correct statistical analysis used to support conclusions?

-Are there concerns about ethical or regulatory requirements being met?

Reviewer #1: Methods:good

Reviewer #2: Due to the wide diversity of species and clinical presentations but with very few cases in each specific situation, I consider that the most appropriate title should be Clinical diversity and treatment of tegumentary leishmaniasis (there is no enough information on ML to be specifically mentioned in the title). 

47 to 53: definition of CL overlaps with that of MCL: "...A patient was considered to have CL if she/he had: (1) cutaneous and/or mucosal lesions..." and "...Muco-cutaneous leishmaniasis (MCL) refers to the simultaneous presence of both mucosal and skin lesions...". I think CL is only cutaneous lesion (not "and/or mucosal").

77: "..., lesion type (papulo-nodular or dry crust or wet crust)..." Not ulcers? Ulcers are the type of lesions usually treated with local therapies.

Reviewer #3: The objectives are intrinscially related with hypothesis. The study design is appropriate to descritive of cohort study proposed. The population is clearly described, however the sample size is not sufficient to power of the hyphotesis. For example, to estimate the prevalence or frequence of mucosal leishmaniasis, or evaluate the response to local therapy the numeber need to be higher. 

Statistical analysis support the conclusions.

There was compliance with ethical precepts. All regulations were followed.

Reviewer #4: Authors have made all the modifications requested and they also have offered the proper clarifications. Methods are adequate.

**Results**

-Does the analysis presented match the analysis plan?

-Are the results clearly and completely presented?

-Are the figures (Tables, Images) of sufficient quality for clarity?

Reviewer #1: REsults: good

Reviewer #2: 111: "... and a previous history of leishmaniasis was reported by 8% of patients in the cohort". I insist that it would be important to know if these patients had had skin or mucosal lesions as a previous episode.

131: "...Excluding rare species and incomplete species (supplementary table 6), we focused on the 5 most frequent infecting complex species...". Good decision.

197, Table 4: "Wash lesion and wound dressing" it could be apparently enough to cure OW CL at similar rates to systemic or topical therapies; so this local care could be recommended as treatment at least to be tested in a clinical trial? It could be THE treatment of many patients in the old world and some in the new world? Or, in the opinion of the authors, are these local measures an adjunctive management of a local or systemic treatment with drugs?

Reviewer #3: The results are clearly presented, however some points in the table 1 (for example) is not complete.

Reviewer #4: Authors have made all the modifications requested an.d they also have offered the proper clarifications. The results section is adequate.

**Conclusions**

-Are the conclusions supported by the data presented?

-Are the limitations of analysis clearly described?

-Do the authors discuss how these data can be helpful to advance our understanding of the topic under study?

-Is public health relevance addressed?

Reviewer #1: conclusions: Good

Reviewer #2: If immunocompromised patients make mucosal lesions in the old world is understandable but that new world patients with all their immune system functioning make recent or late mucosal lesions is more worrying. In fact, what we observe is that mucosal patients in the New World improve with treatment but have frequent recurrences. 

Infiltration and ulceration in tegumentary leishmaniasis is the consequence of an active host immune system fighting against parasites. Accordingly, how can mucosal presentation be explained in the immunosuppressed from the old world compared to the immunocompetent from the new world? Would the authors write a paragraph considering this aspect?

 "...the risk of mucosal involvement is not limited to travels in the New World and effective treatments of CL are not limited to systemic therapy". I fully agree that these are the main findings of this paper, as the authors say in the added paragraph in which they mention the limitations of the study.

Reviewer #3: The conclusions are supported by the results and the limitations are described. 

The manuscript present relevant data for public health

Reviewer #4: Authors have made all the modifications requested and they also offered the proper clarifications. The discussion and conclusions sections are adequate.

**Editorial and Data Presentation Modifications?**

Reviewer #1: (No Response)

Reviewer #2: With the modifications and adjustments in tables and texts it becomes clearer

Reviewer #3: It is necessary rewrite the table 1, containing information about nodular lymphangitis

Reviewer #4: None

**Summary and General Comments**

Reviewer #1: Well revised

Reviewer #2: Paper improved with changes in text and tables. The limitations are now more evident to the reader and allow him/her to read with a better perspective.

Reviewer #3: The authors present comparing tegumentary leishmaniasis from the Old and New World. The topic is quite interesting and current and really requires more information in the literature to support decision-making in public health. However some point need to be improved. Bellow I send my considerations and suggestions. 

1. Abstract

# In table 1, the authors present data on the size of the lesion in mm. I suggest keeping the same unit as shown in table 1 (mm) or leaving cm in the table.

# The authors refer that in the old world there was 5% of mucosal involvement, however, in table 1, the prevalence of mucosal damage was between 2 and 2.5%. Please review this information. It is unclear.

2. Mehtodology

#It is not clear whether all patients with skin lesions were evaluated for the presence of mucosal lesions, as there may be concomitant lesions. If they were evaluated, how was this evaluation done. This information can vary the frequency of mucous involvement, described in this manuscript.

#The authors report a low association with immunodeficiency, however not all patients were tested for HIV. How to state this? 

It would be interesting to describe how many were tested for HIV, to try to make a prevalence of co-infection in those who were tested.

#Regarding the methodology for identifying Leishmania species, was there a standard methodology used in all countries or did each center have its own methodology?

3. Results

# In table 1 there is an asterisk referring to lymphangitis, however there is no lymphangitis data in the table. In the text, the authors described nodular lymphangitis (30% and 6%). Please review this data.

#Out of 459 patients, only 198 had identification of the Leishmania species. In the others it was not tried to identify or was it not possible to identify? Please make this information as clear as possible.

#In table 2, the authors list the Leishmania species identified, with different variables. Regarding the New World, they describe 12 from Peru, as L. braziliensis, and 5 from French Guiana. Regarding L. guyanensis, the authors describe 8 samples from Costa Rica and 7 from French Guiana. Were the other 25 samples from Peru, 20 from Costa Rica and 29 from French Guiana not identified? This information is essential, as in French Guiana there is a prevalence of L. guyanensis and in Costa Rica there is L. infantum causing atypical cutaneous leishmaniasis.

#In table 3, the authors compare the number of lesions between mucous involvement and without mucosal involvement. Was this analysis performed only on patients who had cutaneous and mucosal lesions? Please make this information clearer.

#In table 4, the authors compare local and systemic treatment in old world patients. However, I believe that there may be a bias, since the indication for treatment is based exclusively on the characteristics of the lesion and the number of lesions.

#Regarding treatment, it would also be interesting to analyze the therapeutic response to the species, as some species are known to respond better to certain drugs than others.

Discussion

#From 216 to 220 pages the author mentioned "In particular, no mucosal involvement was observed in patients infected with either L. major or L. tropica, whereas it affected 6% of patients infected with L. braziliensis or L. guyanensis complex (2 MCL, 1 ML) and 22% of patients infected with L. infantum (4 ML, 4 MCL), in whom this complication was strongly (though not exclusively) related to pre-existing immunosuppression."

 I suggest review this affirmation. I believe there is a bias here, as patients who came from the New World had skin lesions. There is no reference for evaluating mucosal lesions in these patients. Furthermore, the mucosal form in the new world usually occurs after years of primary infection, mainly caused by L. braziliensis.

Reviewer #4: None

PLOS authors have the option to publish the peer review history of their article (what does this mean?). If published, this will include your full peer review and any attached files.

Reviewer #1: No

Reviewer #2: Yes: jaime soto

Reviewer #3: No

Reviewer #4: Yes: Gustavo Romero

Figure Files:

Data Requirements:

Reproducibility:

References

---

## [Editor Report · Decision Letter 2]

28 Sep 2021

Dear Mr. Guery,

We are pleased to inform you that your manuscript 'Clinical Diversity and Treatment Results in Tegumentary Leishmaniasis: a European clinical report in 459 patients.' has been provisionally accepted for publication in PLOS Neglected Tropical Diseases.

Best regards,

Edgar M Carvalho

Deputy Editor

Edgar Carvalho

Deputy Editor

---

## [Editor Report · Acceptance letter]

8 Oct 2021

Dear Mr. Guery,

We are delighted to inform you that your manuscript, "Clinical Diversity and Treatment Results in Tegumentary Leishmaniasis: a European clinical report in 459 patients.," has been formally accepted for publication in PLOS Neglected Tropical Diseases.

Best regards,

Shaden Kamhawi

co-Editor-in-Chief

Paul Brindley

co-Editor-in-Chief
